# The Risk of Herpes Zoster in Women with Polycystic Ovary Syndrome: A Retrospective Population-Based Study

**DOI:** 10.3390/ijerph19053094

**Published:** 2022-03-06

**Authors:** Wen-Che Hsieh, Chia-Hung Chen, Yung-Chi Cheng, Teng-Shun Yu, Chung Y. Hsu, Der-Shin Ke, Chih-Ming Lin, Chao-Yu Hsu

**Affiliations:** 1Department of Chinese Medicine, Ditmanson Medical Foundation, Chia-Yi Christian Hospital, Chia-Yi 600, Taiwan; 06085@cych.org.tw; 2Department of Medical Education, Ditmanson Medical Foundation, Chia-Yi Christian Hospital, Chia-Yi 600, Taiwan; 05564@cych.org.tw (C.-H.C.); 11498@cych.org.tw (Y.-C.C.); 3Department of Medical Imaging, Ditmanson Medical Foundation, Chia-Yi Christian Hospital, Chia-Yi 600, Taiwan; 4Department of Rehabilitation, Ditmanson Medical Foundation Chia-Yi Christian Hospital, Chia-Yi 600, Taiwan; 5Management Office for Health Data, China Medical University Hospital, Taichung 404, Taiwan; A85710@mail.cmuh.org.tw; 6College of Medicine, China Medical University, Taichung 404, Taiwan; 7Graduate Institute of Biomedical Sciences, China Medical University, Taichung 404, Taiwan; hsuc@mail.cmuh.org.tw; 8Department of Neurology, Ditmanson Medical Foundation, Chia-Yi Christian Hospital, Chia-Yi 600, Taiwan; 07238@cych.org.tw; 9Department of Laboratory Medicine, Ditmanson Medical Foundation, Chia-Yi Christian Hospital, Chia-Yi 600, Taiwan; 10Department of Optometry/Medical Imaging and Radiological Sciences, Central Taiwan University of Science and Technology, Taichung 406, Taiwan; 11Center for General Education, National Taichung University of Science and Technology, Taichung 404, Taiwan; 12Department of General Education, National Chin-Yi University of Technology, Taichung 411, Taiwan

**Keywords:** polycystic ovary syndrome, herpes zoster, depression

## Abstract

Background: The association between polycystic ovary syndrome (PCOS) and the risk of herpes zoster (HZ) remains unclear. This study investigated the risk of HZ in women with PCOS. Methods: This study used data from the Longitudinal Generation Tracking Database (LGTD 2005) which contains the information of 2 million randomly selected from National Health Insurance beneficiaries. Patients who received a diagnosis of PCOS between 2000 and 2017 were included in the PCOS cohort. Patients who were not diagnosed as having PCOS were randomly selected from the LGTD 2005 and included in the control cohort. Patients who were aged <20 years and had a history of HZ before the index date were excluded. Patients who were in both the cohorts were matched at a ratio of 1:1 through propensity score matching based on age, comorbidities, and medication. The primary outcome was the diagnosis of HZ. Results: A total of 20,142 patients were included in each case and control cohorts. The incidence rates of HZ in the PCOS and control cohorts were 3.92 and 3.17 per 1000 person-years, respectively. The PCOS cohort had a significantly higher risk of HZ than did the control cohort (adjusted hazard ratios [aHR] = 1.26). Among the patients aged 30–39 years, those with PCOS had a significantly higher risk of HZ than did those without PCOS (aHR = 1.31). Among the patients without any comorbidities, those with PCOS had a significantly higher risk of HZ (aHR = 1.26) than did those without PCOS. Conclusion: PCOS is associated with the risk of HZ, especially in young women. The risk of HZ should be addressed while treating patients with PCOS. An HZ vaccine is recommended for these patients.

## 1. Introduction

Polycystic ovary syndrome (PCOS) is caused by an imbalance of female hormones. Patients with PCOS commonly have a high androgen levels and experience irregular menstruation. PCOS has been associated with several problems, such as diabetes, obesity, acne, hirsutism, and infertility. PCOS is commonly diagnosed using the National Institutes of Health (NIH) criteria (defined in 1990), the Rotterdam criteria (established in 2003), and the Androgen Excess and PCOS Society (AE-PCOS) criteria (defined in 2006). A study reported that the prevalence of PCOS diagnosed using the NIH, Rotterdam and AE-PCOS diagnostic criteria were 6%, 10%, and 10%, respectively [1]. Subsequently, Skiba et al. reported that these prevalence rates were 7%, 12%, and 10%, respectively [2]. In a meta-analysis, Naz et al. demonstrated the prevalence of PCOS diagnosed using the NIH, Rotterdam, and AE-PCOS diagnostic criteria in an adolescent population was 3.39%, 11.04%, and 8.03%, respectively [3]. A systematic review and meta-analysis reported that PCOS increases the risk of depression. Cooney et al. reviewed 18 studies and indicated that patients with PCOS had an increased risk of depressive symptoms, with an odds ratio (OR) of 3.78. Another review of 11 studies reported that the OR of the risk of moderate or severe depressive symptoms was 4.18, and the authors recommended screening for depression for patients with PCOS [4].

Herpes zoster (HZ) is a viral infection caused by the reactivation of the varicella-zoster virus (VZV) during the poor immune status of the affected individual. The incidence of HZ ranges from 3 to 5 per 1000 person-years, and the mortality rate of HZ is between 0.017 and 0.465 per 100,000 person-years [5]. Kawai et al. reported that HZ is a global health problem, and its prevalence is expected to increase in the near future with the aging of the world’s population. The HZ vaccine may be administrated to high-risk individuals [5]. Stress caused by physical illness may induce the reactivation of VZV leading to HZ [6,7,8,9,10,11,12,13]. Liao et al. and Choi et al. have reported that psychological diseases, such as depression, also increase the risk of HZ [14,15].

The association between PCOS and HZ risk remains unclear. PCOS and its related syndromes may be stressful for affected individuals. This study investigated the risk of HZ in women with PCOS.

## 2. Materials and Methods

### 2.1. Data Source

The Taiwanese government launched the National Health Insurance (NHI) program in 1995. The NHI program covers more than 99% of the residents of Taiwan. This retrospective study used data from the Longitudinal Generation Tracking Database (LGTD 2005), which contains the information of 2 million randomly selected NHI beneficiaries. From the LGTD 2005, we identified patients with PCOS or HZ using the International Classification of Diseases, Ninth Revision, Clinical Modification (ICD-9-CM), and the International Classification of Diseases, Tenth Revision, Clinical Modification (ICD-10-CM) codes. This study was approved by the Institutional Review Board of China Medical University Hospital Research Ethics Committee (CMUH109-REC2-031(CR-1)).

### 2.2. Study Population

Women who received a diagnosis of PCOS (ICD-9-CM: 256.4, ICD-10-CM: E28.2) between 2000 and 2017 were included in the PCOS cohort. The index date was defined as the first date of PCOS diagnosis. Women who did not receive a diagnosis of PCOS were randomly selected from the LGTD 2005 and included in the control cohort. For the control cohort, a random date between 2000 and 2017 was selected as the index date. The patients in these two cohorts were matched at a ratio of 1:1 using propensity score matching based on age, comorbidities, and medication. According to Ságodi and Barkai, irregular menstruation, anovulatory cycles, and acne are not uncommon among teens, and it can be difficult to distinguish normal teens from those with PCOS [16]. Furthermore, Weinmann et al. reported that the crude HZ incidence for children was only 74 per 100,000 person years over a 12-year period in the varicella vaccine era [17]. Therefore, patients who were aged <20 years or had a history of HZ before the index date were excluded from this study.

### 2.3. Main Outcome and Relevant Variables

The primary outcome was a diagnosis of HZ (ICD-9-CM: 053; ICD-10-CM: B02). We analyzed the following comorbidities considered to be associated with HZ: diabetes (ICD-9-CM: 250; ICD-10-CM: E08-E13), chronic kidney disease (ICD-9-CM: 585; ICD-10-CM: N18), coronary artery disease (ICD-9-CM: 410-414; ICD-10-CM: I20-I22, I24, I25), cancer (ICD-9-CM: 140-208; ICD-10-CM: C00-C26, C30-C34, C37-C41, C43-C50, C53-C55, C4A, C7A, D03, Z51.12), obesity (ICD-9-CM: 278; ICD-10-CM: E66), and depression (ICD-9-CM: 296, 298, 300, 301, 311; ICD-10-CM: F32, F33, F34, F41). In addition, the use of medications, namely Metformin, Rosiglitazone, and Spironolactone, was examined.

### 2.4. Statistical Analysis

Differences between continuous variables were analyzed using the Student’s *t*-test. The chi-square test was used to determine differences in categorical variables. We estimated the risk of HZ in the PCOS and control cohorts by calculating adjusted hazard ratios (aHR) and 95% confidence intervals (CI) by using univariate and multivariate Cox proportional hazards regression models. Multivariate models were adjusted for age, comorbidities, and medications. The Kaplan–Meier survival curve was plotted to compare the cumulative incidence between the cohorts, and the log-rank test was used to examine the differences. All statistical analyses were performed using SAS version 9.4 (SAS Institute, Inc., Cary, NC, USA), and all graphs were plotted using R studio (3.5.2).

## 3. Results

A total of 20,142 patients each were included in the PCOS and control cohorts, respectively. Table 1 lists the patient characteristics of the two cohorts. The mean (standard deviation) ages of the patients in the control and PCOS cohorts were 29.36 (6.43) and 29.20 (6.24) years, respectively. After propensity score matching, the prevalence of comorbidities, namely diabetes, chronic kidney disease, coronary artery disease, cancer, obesity, and depression, did not differ between the PCOS and control cohorts. In addition, the use of Metformin, Rosiglitazone, and Spironolactone did not differ between the cohorts.

The incidence of HZ in the PCOS and control cohorts was 3.92 and 3.17 per 1000 person years, respectively (Table 2). The incidence of HZ was higher in the PCOS cohort than in the control cohort (log-rank test, *p* < 0.001; Figure 1). In addition, the PCOS cohort had a significantly higher risk of HZ than did the control cohort (aHR = 1.26, 95% CI = 1.12–41). Those aged 30–39 years (aHR = 1.15, 95% CI = 1.02–1.30) and 40–65 years (aHR = 1.90, 95% CI = 1.57–2.30) had a significantly higher risk of HZ than did those aged 20–29 years.

In the age group 30–39 years, patients with PCOS had a significantly higher risk of HZ than those without PCOS (aHR = 1.31, 95% CI = 1.09–1.58) (Table 3). Among the patients with depression, those with PCOS had a higher risk of HZ than those without PCOS (aHR = 1.44, 95% CI = 1.10–1.89). Among the patients without depression, those with PCOS still had a higher risk of HZ than those without PCOS (aHR = 1.22, 95% CI = 1.08–1.39). Overall, among the patients without any comorbidities, the patients with PCOS had a significantly higher risk of HZ (aHR = 1.26, 95% CI = 1.11–1.44) (Table 3).

## 4. Discussion

This is the first study to investigate the association between PCOS and HZ risk. We observed that the women with PCOS were 1.26 times more likely to have HZ. Among the patients without any comorbidities, those with PCOS had a significantly higher risk of HZ (aHR = 1.26). The risk of HZ should not be ignored in patients with PCOS.

Diabetes, obesity, acne, hirsutism, and infertility are commonly observed in patients with PCOS, and these conditions are associated with depression. One study reported that patients with diabetes were twice as likely as those without diabetes to develop depression [18]. The risk of depression was higher in patients with type 1 and type 2 diabetes. Roy and Lloyd performed a systemic review and observed that the prevalence of depression was approximately three times and two times higher in patients with type 1 (12% vs. 3.2%) and type 2 diabetes (19.1% vs. 10.7%), respectively [19]. In addition, the risk of depression increased with the severity of diabetic complications [20].

Obesity is associated with an increased risk of depression. Luppino et al. reported that the odds ratio (OR) of depression risk was 1.55 in individuals with obesity [21]. Sutaria et al. indicated that female children with obesity had a significantly higher risk of depression (OR = 1.44) than normal-weight female children. However, this association was not observed in male children with obesity. The authors suggested screening for depression in children with obesity, particularly in female children [22].

Yang et al. reported that the prevalence of major depression was higher in patients with acne than in those without acne (0.77% vs. 0.56%, *p* < 0.0001). They also reported that women with acne had a significantly higher risk of major depression. The ORs of major depression risk for women with and without acne were 2.78 and 1.85, respectively [23]. The risk of depression was high in patients with acne, Samuels et al. suggested aggressive acne treatment and psychiatric screening for these patients [24].

The Ferriman-Gallwey (F-G) scale is a scoring system used to determine hair growth. Ekbäck et al. investigated the correlation between hirsutism (examined by the F-G scale) and depression (examined by Hospital Anxiety and Depression Scale [HADS]) using a self-administered questionnaire survey. The authors reported that the rates of mild (HADS-D = 8–10) and moderate or severe (HADS-D > 10) depression were 48% and 16%, respectively, in 127 women with excessive hair growth. In addition, the hair growth level was significantly associated with depression [25]. The association between the level of hair growth and the severity of depression was confirmed.

Al-Homaidan reported that 53.8% of women with infertility had depression. The author observed that the mean Beck Depression Inventory score was significantly higher in women with infertility than in those without infertility (12.23 vs. 8.31, *p* < 0.001) [26]. Ramezanzadeh et al. reported that depression was commonly observed after 4–6 years of infertility among women, and severe depression was noted after 7–9 years of infertility. The authors suggested that the mental health of women with infertility should be managed properly [27].

Two nationwide cohort studies have examined the association between depression and HZ. Liao et al. reported that the incidence of HZ was significantly higher in patients with depression than in those without depression (4.58 vs. 3.54 per 1000 person-years). Compared with patients without depression, patients with depression had a HR of 1.11 [14]. Choi et al. reported that the infection rates of HZ were 6.8% and 6.3% (*p* < 0.001) in patients with and without depression, respectively, and the HR was 1.09, particularly in women with depression aged <60 years [15]. The risk of HZ in patients with depression is well reported. In our study, among patients with depression, those with PCOS had a significantly higher risk of HZ than did those without PCOS (aHR = 1.44; Table 3). The related syndromes of PCOS are associated with depression and this depression is thought to cause HZ.

A previous study found that VZV can be reactivated after stress by checking VZV DNA of saliva samples from astronauts. Mehta et al. reported only one sample was positive for VZV DNA before space flight; however, 30% saliva samples were positive during and after space flight [28]. Recently, Rooney et al. reported that VZV shedding from astronauts’ saliva or urine samples was 41% in space shuttle and increased to 65% in international space station. Stress hormones were increased during spaceflight due to activation of hypothalamic-pituitary-adrenal and sympathetic-adrenal-medullary axes. It may influence the cell mediated immunity and lead to reactivation of HZ [29]. In our study, among the patients without depression, those with PCOS still had a significantly higher risk of HZ than did those without PCOS (aHR = 1.22). These results indicate that that PCOS itself is a stressful factor for individuals. Among the patients without any comorbidities, those with PCOS had a higher risk of HZ than did those without PCOS (aHR = 1.26, 95% CI = 1.11–1.44). In the groups without medication, patients with PCOS showed a higher rate of HZ occurrence than those without PCOS. The burden of PCOS can cause stress, leading to the development of HZ.

This retrospective study has several limitations. First, because the diagnosis of PCOS or HZ was established by different specialists, diagnostic bias may occur. According to the rules of the Taiwan NHI Administration, all insurance claims should be sent to the NHI Administration after clinical evaluations. To prevent unnecessary or inappropriate prescriptions, the NHI Administration has a penalty system that involves a strict review by experts. Therefore, the diagnostic codes are reliable. Second, the severity of PCOS may affect outcomes. However, information regarding the severity of diseases was not available in this study because of insufficient details regarding ICD-9-CM codes. Third, smoking status is not provided in the NHI research database. One study reported that former smokers were 1.17 times more likely to develop HZ than those who had never smoked [30]. Thus, outcomes may have been overestimated or underestimated in this study. Fourth, self-payment for treatment is not recorded in the NHI research database. Details regarding self-payment for the HZ vaccine or pain control through acupuncture are not available in the NHI research database. These treatments may affect the outcomes of HZ. Because this is a retrospective cohort study, temporality between PCOS diagnosis and HZ risk might exist; therefore, causal associations can be present. However, the causality should be determined in subsequent studies. Despite some limitations, this evidence-based, nationwide study including a large sample size can set the basis for additional studies on population medicine.

## 5. Conclusions

PCOS is associated with the risk of HZ, especially in young women. The risk of HZ should be addressed while treating patients with PCOS. An HZ vaccine is recommended for these patients.

## Figures and Tables

**Figure 1 ijerph-19-03094-f001:**
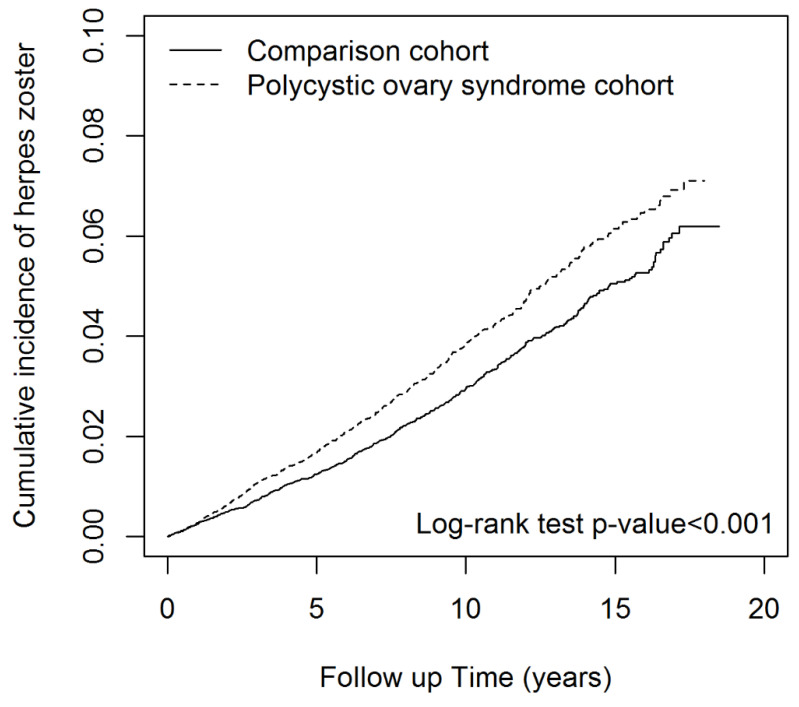
Cumulative incidence of herpes zoster in patients with polycystic ovary syndrome compared to controls.

**Table 1 ijerph-19-03094-t001:** The characteristics of patients with or without polycystic ovary syndrome.

	Polycystic Ovary Syndrome				
	No (*n* = 20,142)		Yes (*n* = 20,142)		
Variable	*n*	%	*n*	%	*p*-Value
Age (year)					0.9355
20–29	11,247	55.84	11272	55.96	
30–39	7531	37.39	7518	37.32	
40–65	1364	6.77	1352	6.71	
Age mean (SD)	29.36	(6.43)	29.2	(6.24)	
Comorbidities					
Diabetes	582	2.89	576	2.86	0.8580
Chronic kidney disease	39	0.19	41	0.20	0.8229
Coronary artery disease	325	1.61	321	1.59	0.8739
Cancer	301	1.49	307	1.52	0.8063
Obesity	647	3.21	628	3.12	0.5887
Depression	3244	16.11	3226	16.02	0.8070
Medication					
Metformin	3118	15.48	3107	15.43	0.8795
Rosiglitazone	69	0.34	70	0.35	0.9323
Spironolactone	646	3.21	655	3.25	0.7998

**Table 2 ijerph-19-03094-t002:** Cox model measured hazard ratios and 95% confidence intervals of herpes zoster associated with and without polycystic ovary syndrome and covariates.

	Herpes Zoster			Crude			Adjusted		
Characteristics	Event	Pearson-Years	IR	HR	(95% CI)	*p*-Value	HR	(95% CI)	*p*-Value
Polycystic ovary syndrome									
No	575	181,597	3.17	1.00	(reference)	-	1.00	(reference)	-
Yes	632	161,415	3.92	1.26	(1.12, 1.41) ***	<0.001	1.26	(1.12, 1.41) ***	<0.001
Age (year)									
20–29	621	199,394	3.11	1.00	(reference)	-	1.00	(reference)	-
30–39	448	122,351	3.66	1.19	(1.06, 1.35) **	0.0046	1.15	(1.02, 1.30) *	0.0249
40–65	138	21,267	6.49	2.13	(1.77, 2.56) ***	<0.001	1.90	(1.57, 2.30) ***	<0.001

IR: Incidence rate, per 1000 persons/years; HR: Hazard ratio; CI: confidence interval; Adjusted HR: adjusted for age, comorbidities, and medications in Cox proportional hazards regression. * *p* < 0.05, ** *p* < 0.01, *** *p* < 0.001.

**Table 3 ijerph-19-03094-t003:** Cox model with hazard ratios and 95% confidence intervals of herpes zoster associated patients stratified by polycystic ovary syndrome.

9	Polycystic Ovary Syndrome											
	No			Yes								
Variable	Event	Pearson Year	IR	Events	Pearson Year	IR	cHR	(95% CI)	*p*-Value	aHR	(95% CI)	*p*-Value
Age, year												
20–29	305	105,261	2.90	316	94,132	3.36	1.18	(1, 1.38) *	0.0442	1.17	(1.00, 1.37)	0.0542
30–39	209	65,208	3.21	239	57,143	4.18	1.33	(1.11, 1.61) **	0.0025	1.31	(1.09, 1.58) **	0.0046
40–65	61	11,128	5.48	77	10,139	7.59	1.43	(1.02, 2) *	0.0369	1.40	(1.00, 1.96)	0.0500
Comorbidity ^§^												
No	437	147,787	2.96	468	128,741	3.64	1.25	(1.1, 1.43) ***	<0.001	1.26	(1.11, 1.44) ***	<0.001
Yes	138	33,809	4.08	164	32673	5.02	1.24	(0.99, 1.55)	0.065	1.24	(0.99, 1.56)	0.0622
Diabetes												
No	554	177,205	3.13	610	156,901	3.89	1.27	(1.13, 1.42) ***	<0.001	1.26	(1.13, 1.42) ***	<0.001
Yes	21	4391	4.78	22	4513	4.87	1.02	(0.56, 1.86)	0.9365	1.02	(0.56, 1.88)	0.9403
Coronary artery disease												
No	566	179,102	3.16	615	159,146	3.86	1.24	(1.11, 1.4) ***	<0.001	1.24	(1.11, 1.39) ***	<0.001
Yes	9	2495	3.61	17	2269	7.49	2.07	(0.92, 4.64)	0.0782	2.21	(0.98, 5.01)	0.0569
Cancer												
No	547	178,675	3.06	610	158,265	3.85	1.28	(1.14, 1.44) ***	<0.001	1.29	(1.15, 1.45) ***	<0.001
Yes	28	2921	9.58	22	3149	6.99	0.73	(0.42, 1.28)	0.274	0.8	(0.45, 1.41)	0.4337
Obesity												
No	557	177,180	3.14	612	157,044	3.90	1.26	(1.12, 1.41) ***	<0.001	1.26	(1.12, 1.41) ***	<0.001
Yes	18	4417	4.08	20	4371	4.58	1.13	(0.60, 2.15)	0.6972	1.09	(0.57, 2.07)	0.7977
Depression												
No	483	157,300	3.07	509	138,280	3.68	1.22	(1.08, 1.38) **	0.0017	1.22	(1.08, 1.39) **	0.0016
Yes	92	24,297	3.79	123	23,135	5.32	1.42	(1.08, 1.86) *	0.0108	1.44	(1.10, 1.89) **	0.0082
Medication												
Metformin												
No	456	145,757	3.13	523	133,614	3.91	1.27	(1.12, 1.44) ***	<0.001	1.26	(1.11, 1.43) ***	<0.001
Yes	119	35,840	3.32	109	27,801	3.92	1.22	(0.94, 1.58)	0.1397	1.21	(0.93, 1.57)	0.1617
Rosiglitazone												
No	571	180,783	3.16	628	160,647	3.91	1.26	(1.12, 1.41) ***	<0.001	1.26	(1.12, 1.41) ***	<0.001
Yes	4	813	4.92	4	768	5.21	1.12	(0.28, 4.54)	0.869	1.56	(0.17, 14.25)	0.6923
Spirolactone												
No	547	175,170	3.12	608	155,453	3.91	1.28	(1.14, 1.43) ***	<0.001	1.27	(1.13, 1.43) ***	<0.001
Yes	28	6427	4.36	24	5962	4.03	0.92	(0.53, 1.59)	0.7742	0.95	(0.55, 1.64)	0.8439

IR: Incidence rate, per 1000 persons/years; HR: Hazard ratio; CI: confidence interval; Adjusted HR: adjusted for age, comorbidities, and medications in Cox proportional hazards regression. ^§^ Individuals with any comorbidity of diabetes, coronary artery disease, cancer, obesity, and depression were classified into the comorbidity group. * *p* < 0.05, ** *p* < 0.01, *** *p* < 0.001.

## Data Availability

Data are available from the National Health Insurance Research Database which provided by the Taiwan National Health Insurance Administration. However, data cannot be disclosed due to the law of personal data protection. Data can be requested from Taiwan NHI Administration through a formal application (http://nhird.nhri.org.tw, accessed on 27 December 2021).

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
