# Peer review of "The Risk of Herpes Zoster in Women with Polycystic Ovary Syndrome: A Retrospective Population-Based Study"

_ijerph, 2022, doi:10.3390/ijerph19053094_

Round 1

Reviewer 1 Report

Comments from reviewer.

The article is well written.

The following minor change may make the article better.

Page 1, Abstract, line 36

Page 3 Materials and Methods line 98

“Patients who were aged <20 years and had a history of HZ before the index date were excluded.”

The reviewer could not understand why the patients under 20 years have to be excluded. It may need to be explained

Page 6 Discussion line 175.

Luppino et al. reported that the OR of depression risk was 1.55 in individuals with obesity 〔19〕.

The reviewer could not find the formal name of “OR”. It may need to be indicated by formal name.

Page 8 Conclusion line246-247,

It might be better to add a simple explanation about the mechanism between the risk of HZ and PCOS, for example, “The related syndromes of PCOS are associated with depression and this depression is thought to cause HZ.”, in order to make readers understand the contents of the article.

Author Response

The article is well written.

The following minor change may make the article better.

Page 1, Abstract, line 36

Page 3 Materials and Methods line 98

“Patients who were aged <20 years and had a history of HZ before the index date were excluded.”

The reviewer could not understand why the patients under 20 years have to be excluded. It may need to be explained

Answer: Thanks for your comment. We added several sentences in the text.

“According to the report of Ságodi & Barkai, irregular menstruation, anovulatory cycles, and acne are not uncommon among teens, and it can be difficult to distinguish normal teens from those with PCOS.  Furthermore, Weinmann et al. reported that the crude HZ incidence for children was only 74 per 100,000 person years over a 12-year period in the varicella vaccine era. Therefore, patients who were aged <20 years or had a history of HZ before the index date were excluded from this study.”

(Ságodi L, Barkai L. Diagnostic difficulties of polycystic ovarian syndrome in adolescent girls. Orv Hetil. 2013 27; 154(4): 136-142.)

(Weinmann S, Naleway AL, Koppolu P, Baxter R, Belongia EA, Hambidge SJ, Irving SA, Jackson ML, Klein NP, Lewin B, Liles E, Marin M, Smith N, Weintraub E, Chun C. Incidence of herpes zoster among children: 2003-2014. Pediatrics. 2019 Jul;144(1):e20182917.)

Page 6 Discussion line 175.

Luppino et al. reported that the OR of depression risk was 1.55 in individuals with obesity 〔19〕.

The reviewer could not find the formal name of “OR”. It may need to be indicated by formal name.

Answer: Thanks for your comment. We added “odds ratio” in the text.

Page 8 Conclusion line246-247,

It might be better to add a simple explanation about the mechanism between the risk of HZ and PCOS, for example, “The related syndromes of PCOS are associated with depression and this depression is thought to cause HZ.”, in order to make readers understand the contents of the article.

Answer: Thanks for your comment. We added the sentence “The related syndromes of PCOS are associated with depression and this depression is thought to cause HZ.” in the text. And we erased several sentences to make the clear results.

Reviewer 2 Report

Authors reported that patients with PCOS had higher risk of HZ regardless of any comorbidities with large cohort study. This report has novel findings. But it should be much simplified.

  1. Discussion: Too long. It should be described only relationship between PCOS and HZ.
  2. Please explain why patients who were aged <20 years were excluded in this study.
  3. Line 78 and 222: Authors indicated PCOS is stressful. Please provide evidence that PCOS is stressful by your data or other previous manuscripts.

    And please provide the evidence of relationship between stress and HZ.

  4. Table 2: HZ incidence was analyzed by age and comorbidities. But these data indicate general incidence of HZ because of including HZ patients with PCOS and without PCOS. It should be deleted.
  5. Table 3: Please discuss the reasons why patients with PCOS showed higher rate of HZ than those without PCOS in the groups without medication.

Author Response

Authors reported that patients with PCOS had higher risk of HZ regardless of any comorbidities with large cohort study. This report has novel findings. But it should be much simplified.

  1. Discussion: Too long. It should be described only relationship between PCOS and HZ.

Answer: Thanks for your comment. In the section of “Discussion”, we discuss the PCOS related syndromes such as diabetes, obesity, acne, hirsutism and infertility, and all of them were associated with depression. And, depression is associated with HZ. We explained the probable mechanism of PCOS related syndromes-depression-herpes zoster. We corrected and deleted several sentences, and shortened the section of “Discussion”.

  1. Please explain why patients who were aged <20 years were excluded in this study.

Answer: Thanks for your comment. We added several sentences in the text.

“According to the report of Ságodi & Barkai, irregular menstruation, anovulatory cycles, and acne are not uncommon among teens, and it can be difficult to distinguish normal teens from those with PCOS.  Furthermore, Weinmann et al. reported that the crude HZ incidence for children was only 74 per 100,000 person years over a 12-year period in the varicella vaccine era. Therefore, patients who were aged <20 years or had a history of HZ before the index date were excluded from this study.”

(Ref: Ságodi L, Barkai L. Diagnostic difficulties of polycystic ovarian syndrome in adolescent girls. Orv Hetil. 2013 27; 154(4): 136-142.)

(Ref: Weinmann S, Naleway AL, Koppolu P, Baxter R, Belongia EA, Hambidge SJ, Irving SA, Jackson ML, Klein NP, Lewin B, Liles E, Marin M, Smith N, Weintraub E, Chun C. Incidence of herpes zoster among children: 2003-2014. Pediatrics. 2019 Jul;144(1):e20182917.)

  1. Line 78 and 222: Authors indicated PCOS is stressful. Please provide evidence that PCOS is stressful by your data or other previous manuscripts. And please provide the evidence of relationship between stress and HZ.

Answer: Thanks for your comment. We added several sentences to discuss the relationship between stress and HZ. “A previous study found that VZV can be reactivated after stress by checking VZV DNA of saliva samples from astronauts. Mehta et al. reported only one sample was positive for VZV DNA before space flight; however, 30% saliva samples were positive during and after space flight. Recently, Rooney et al. reported that VZV shedding from astronauts’ saliva or urine samples was 41% in space shuttle and increased to 65% in international space station. Stress hormones were increased during spaceflight due to activation of hypothalamic-pituitary-adrenal and sympathetic-adrenal-medullary axes. It may influence the cell mediated immunity and lead to reactivation of HZ.”

(Ref: Mehta SK, Cohrs RJ, Forghani B, Zerbe G, Gilden DH, Pierson DL. Stress-induced subclinical reactivation of varicella zoster virus in astronauts. J Med Virol. 2004 Jan; 72(1): 174-179.)

(Ref: Rooney BV, Crucian BE, Pierson DL, Laudenslager ML, Mehta SK. Herpes virus reactivation in astronauts during spaceflight and its application on earth. Front Microbiol. 2019 Feb 7;10:16.)

  1. Table 2: HZ incidence was analyzed by age and comorbidities. But these data indicate general incidence of HZ because of including HZ patients with PCOS and without PCOS. It should be deleted.

Answer: Thanks for your comment. We deleted the part of “comorbidities” in the Table 2. Only keep the age and incidence to exhibit the incidence of HZ was higher in the PCOS cohort than in the control cohort, the PCOS cohort had a significantly higher risk of HZ than did the control cohort. And we also deleted several sentences in the section of “Results”.

  1. Table 3: Please discuss the reasons why patients with PCOS showed higher rate of HZ than those without PCOS in the groups without medication.

Answer: Thanks for your comment. Medication can improve the condition of PCOS and may influence the occurrence of HZ. There was no difference between the patients with and without PCOS in the group of medication. In the groups without medication, patients with PCOS showed a higher rate of HZ occurrence than those without PCOS. It means that the burden of PCOS can cause stress, leading to the development of HZ. We added a sentence in the section of “Discussion”. This probably further supports the association between PCOS and development of HZ.

Round 2

Reviewer 2 Report

Thank you for modifying manuscript. Now, it is acceptable.